# Influence of Previous COVID-19 and Mastitis Infections on the Secretion of Brain-Derived Neurotrophic Factor and Nerve Growth Factor in Human Milk

**DOI:** 10.3390/ijms22083846

**Published:** 2021-04-08

**Authors:** Veronique Demers-Mathieu, Dustin J. Hines, Rochelle M. Hines, Sirima Lavangnananda, Shawn Fels, Elena Medo

**Affiliations:** 1Department of Neonatal Immunology and Microbiology, Medolac Laboratories A Public Benefit Corporation, Boulder City, NV 89005, USA; slavangnananda@medolac.com (S.L.); sfels@medolac.com (S.F.); emedo@medolac.com (E.M.); 2Department of Psychology, University of Nevada Las Vegas, Las Vegas, NV 89154, USA; dustin.hines@unlv.edu (D.J.H.); rochelle.hines@unlv.edu (R.M.H.)

**Keywords:** neurodevelopment, breast milk, neurological symptoms, SARS-CoV-2, infectious disease, neurotrophins, central nervous system, newborns, immune system, breastfeeding

## Abstract

Background: Brain-derived neurotrophic factor (BDNF) and nerve growth factor (NGF) play a critical role in neurodevelopment, where breast milk is a significant dietary source. The impact of previous COVID-19 infection and mastitis on the concentration of BDNF and NGF in human milk was investigated. Methods: Concentrations of BDNF and NGF were measured via ELISA in human milk samples collected from 12 mothers with a confirmed COVID-19 PCR, 13 mothers with viral symptoms suggestive of COVID-19, and 22 unexposed mothers (pre-pandemic Ctl-2018). These neurotrophins were also determined in 12 mothers with previous mastitis and 18 mothers without mastitis. Results: The NGF concentration in human milk was lower in the COVID-19 PCR and viral symptoms groups than in the unexposed group, but BDNF did not differ significantly. Within the COVID-19 group, BDNF was higher in mothers who reported headaches or loss of smell/taste when compared with mothers without the respective symptom. BDNF was lower in mothers with mastitis than in mothers without mastitis. Conclusions: Previous COVID-19 and mastitis infections changed differently the secretion of NGF and BDNF in human milk. Whether the changes in NGF and BDNF levels in milk from mothers with infection influence their infant’s development remains to be investigated.

## 1. Introduction

Brain-derived neurotrophic factor (BDNF) plays an essential role in neurodevelopment [1] and regulation of neuronal survival and proliferation [2,3]. BDNF is involved in learning and memory processing (short and long-term), particularly in memory persistence and storage [4], as this neurotrophic protein is active in the hippocampus and cortex. Nerve Growth Factor (NGF) is another neurotrophin (NF) that has a crucial role in the process of neuronal angiogenesis/apoptosis, brain development, and tissue proliferation and differentiation [5].

Breastfed infants are reported to have a higher total behavior rating scale and motor quality percentile rank than formula-fed infants [6]. BDNF levels are positively correlated with the total behavior rating scale (TBRS), which indicates improved cognition. Human milk BDNF could be one of the causative factors of higher cognitive functions in breastfed infants [6]. Serum BDNF levels are higher in mothers than in their infants and lower in preterm infants than in term infants [7]. BDNF levels could represent the maturity of the nervous and immune systems to produce NFs. Umbilical cord BDNF levels increased with increasing gestational age (GA) (24–28 wks < 29–35 wks < 36–40 wks) [8]. These findings underline the differential degree of peripheral and central nervous system (CNS) maturity between preterm and term infants. Preterm birth represents an abrupt cessation of intrauterine growth and maturation, explaining the lower BDNF levels during early extrauterine life. Administration of BDNF has shown to offer significant neuroprotection against delayed (apoptotic) neuronal death following brain injury in the neonate [9,10]. Term and preterm infants possess an immature nervous system, and these NFs may attenuate preterm effects and stimulate neurodevelopment. However, no research has demonstrated the effects of human milk NFs on preterm and term infants’ neurodevelopment.

Previous studies have detected a large difference in BDNF levels in mother’s breast milk (from non-detectable to ~400 pg/mL) [6,7,11,12]. Maternal factors influence the nervous and immune system (including previous diseases and infections, genetic factors, nutrition, and mental status) and change the secretion of NFs in human milk. Lower plasma BDNF levels were associated with increasing severity of depression [13]. In a murine model, lower BDNF level was also positively correlated with anxiety and aggression [14]. The maternal immune system can change the secretions of BDNF by activated human T cells, B cells, and monocytes [10]. Mast cells, monocytes and macrophages, T cells, and B cells express NGF during the inflammatory response [15].

During COVID-19 infection, headache, dizziness, and impaired consciousness are common neurological symptoms observed [16]. These neurological symptoms may be interrelated with the production of NFs in human milk. A recent study reported that serum BDNF levels were lower in patients with severe or moderate COVID-19 disease than those with mild COVID-19 [17]. Low serum BDNF levels were associated with severe SARS-CoV-2 infection. The BDNF levels were restored during the patients’ recovery, but the NGF level was not investigated in this study.

Another relevant maternal infection during breastfeeding is mastitis. Mastitis is caused by milk stasis and is associated with various causes, including maternal stress and fatigue, physical and psychological illnesses in the mother or baby that prevent mothers from feeding or infants from attaching to breasts, poor attachment by infants, damaged nipples colonized by Staphylococcus, and blocked nipple pores or ducts [18,19]. The cellular composition of human milk and local immune defenses within the breast are altered during mastitis [20]. Mastitis is also the primary cause of a reduction in milk production, where 25% of mothers experiencing mammary local inflammation stop breastfeeding due to the pain [21]. The effect of post-mastitis on the secretion of NFs in human milk is still unexplored.

The impact of COVID-19 infection on the concentrations of BDNF and NGF in human milk is still unknown. This study aimed to compare the concentration of BDNF and NGF in human milk between mothers with previous COVID-19 infection and unexposed mothers (control pre-pandemic 2018). This investigation’s clinical importance is that recovered COVID-19 mothers may secrete less human milk NFs than unexposed mothers and thereby influence the brain development of their infants. This study also compared NF levels between mothers with previous mastitis and without mastitis to evaluate the impact of maternal stress and fatigue that mastitis induces during breastfeeding.

## 2. Results

### 2.1. COVID-19 Study

#### 2.1.1. Maternal Demographics

Maternal demographic details of participant groups are presented in Table 1. Postpartum time, infant gender, and maternal age were comparable between the COVID-19 PCR, viral symptoms suggestive of COVID-19, and the unexposed groups. The number of mothers experiencing symptoms that could affect the CNS was reported in Table 1, including headaches, loss of smell and taste, fatigue, and nasal congestion. Individual characteristics of the COVID-19 PCR and viral symptoms suggestive of COVID-19 groups are described in Appendix A.

#### 2.1.2. BDNF and NGF in Milk from COVID-19 and Unexposed Mothers

BDNF alone in human milk did not differ between the three groups (Figure 1A). Concentrations of NGF and cumulative NF (BDNF+NGF) in human milk were lower in the COVID-19 PCR and viral symptoms suggestive of COVID-19 groups than in the unexposed group (*p* < 0.01, Figure 1B,D), but did not differ between COVID-19 PCR and viral symptoms groups. When the two outlier cases in the unexposed group for NGF were removed, NGF and cumulative NF levels were still higher in the unexposed group than in the COVID-19 PCR and viral symptoms groups (*p* < 0.05, Figure 1C,E). NGF concentration was 8.9-fold higher than BDNF concentration in milk from exposed mothers (*p* < 0.001, Figure 1F). NGF and BDNF concentrations were comparable in the COVID-19 PCR and viral symptoms groups (Figure 1F).

#### 2.1.3. Effects of Maternal Factors on Human Milk BDNF and NGF

Of particular interest, BDNF was 20.4-fold higher in mothers with headaches than mothers without headaches in the COVID-19 group (*p* = 0.048, Figure 2A). BDNF was 35-fold higher in mothers with loss of smell and taste than mothers without loss of smell/taste in the COVID-19 group (*p* = 0.003, Figure 2B). The other viral symptoms did not affect the concentration of NFs in human milk.

BDNF and NGF concentrations were not influenced by the elapsed time from infection to milk collection in mothers with a confirmed COVID-19 PCR (Figure 2C,D) and in mothers with viral symptoms suggestive of COVID-19 (Figure 2E,F). BDNF did not correlate with NGF in the COVID-19 PCR group (*p* = 0.75) or in the viral symptoms group (*p* = 0.26). BDNF concentration tended to be positively correlated with NGF concentration in the unexposed group (*p* = 0.080, *r* = 0.38). However, when the two outlier cases in the unexposed group were removed, no correlation was observed between NGF and BDNF levels (*p* = 0.66). Infant gender, infant age, and maternal age did not affect the concentrations of NFs in human milk.

### 2.2. Mastitis Study

#### 2.2.1. Maternal Demographics

Maternal demographic details of participant groups are presented in Table 2. Postpartum time, infant gender, and maternal age were comparable between mothers with mastitis and mothers without mastitis. Mothers with mastitis reported being stressed and have extreme fatigue during breast infection. Mothers without mastitis reported having experienced any infection in the last two years and any intense stress and fatigue during milk collection.

#### 2.2.2. BDNF and NGF in Human Milk between Mastitis and No Mastitis Groups

The concentration of BDNF and cumulative NF (BDNF + NGF) in human milk were 5.0- (*p* = 0.018, Figure 3A) and 3.2-fold (*p* = 0.036, Figure 3C) higher, respectively, in mothers without mastitis than in mothers with previous mastitis, but NGF alone did not differ (Figure 3B). The elapsed time from mastitis to milk collection did not correlate with the concentration of NFs in human milk. BDNF and NGF concentrations were comparable in milk from mothers with or without mastitis (Figure 3D). Infant gender, infant age, and maternal age did not affect the concentrations of NFs in human milk.

## 3. Discussion

BDNF and NGF play an essential role in controlling apoptosis during brain development and enhancing neuronal differentiation, survival, and growth of neurons [22]. Human milk NFs may modulate neurotrophic functions and stimulate neurodevelopment during breastfeeding [23]. Neurodevelopment impairment is less frequent in breastfed infants than in formula-fed infants during the first three weeks of postnatal age [24]. Human milk intake is associated with improved brain development [25] and could influence neurogenesis, neuronal differentiation, myelination, and synaptogenesis in neonatal development [26]. Therefore, the changes in concentrations in human milk NFs when mothers have infections with neurological symptoms could influence their infants’ neurodevelopment.

Mild COVID-19 infection can induce neurological symptoms, including headaches, fatigue, sensations of numbness or tingling, and cognitive difficulties and inflammation of the brain [27,28]. SARS-CoV-2 infected olfactory mucosa (nose) can result in the loss of taste and smell [29]. The virus could then pass into the CNS via axonal transport and exert pathological effects in the CNS [29,30]. Another potential route of viral entry is when leukocytes carry SARS-CoV-2 from the cerebellum across the blood-brain barrier [29]. As NFs are produced by neuronal cells [5] and immune cells [31], their levels could change due to the destruction of neural cells and immune cells during COVID-19 infection. Indeed, serum BDNF levels were lower in patients with severe or moderate COVID-19 infection than those with mild COVID-19 [17]. Therefore, it is feasible that COVID-19 infection could influence BDNF and NGF secretion in human milk.

This study provides the first data on BDNF and NGF concentrations in human milk samples from mothers with a confirmed COVID-19 PCR test, mothers with previous viral symptoms suggestive of COVID-19, and unexposed mothers (control pre-pandemic 2018). We demonstrated that NGF concentrations in human milk were lower in the COVID-19 PCR and viral symptoms groups than in the unexposed group, but BDNF was comparable. The NGF and BDNF concentrations did not differ between COVID-19 PCR and viral symptoms suggestive of COVID-19 groups. These results suggest that SARS-CoV-2 infection influences the secretion of selected NFs (NGF) in human milk. A recent study demonstrated that serum BDNF levels were lower in patients with severe or moderate disease than in patients with mild disease (6.3 vs. 7.4 ng/mL) [17], but their symptoms were not reported. Low serum BDNF levels were associated with severe SARS-CoV-2 infection, lower absolute lymphocyte count, and higher C-reactive protein (a marker of inflammation) [17]. As lymphocytes contribute to the secretion of BDNF, lymphopenia induced a reduction of BDNF secretion in patients with severe COVID-19. The authors also observed that BDNF levels were restored during the patient’s recovery, but the elapsed time from the infection to the collection was not reported [17]. Whether the reduced level of NGF in milk from recovered COVID-19 mothers is associated with a lower population of lymphocytes remains to be evaluated.

We observed that the BDNF level was higher in mothers with headaches or with loss of smell/taste than in mothers without the respective symptom, but NGF levels did not differ according to symptom profile. These findings indicate that specific COVID-19 symptoms related to the CNS (including headaches) changed the secretion of BDNF in human milk. However, these results are from a small number of mothers, and a larger sample size is needed to generalize this phenomenon. Morichi et al. [32] also found that serum BDNF levels increased in children hospitalized with influenza-associated encephalopathy (IAE) compared to control (without viral infection) and respiratory syncytial virus groups, but did not differ with human herpesvirus type 6 and rotavirus-associated encephalopathy. These authors suggested that BDNF might be associated with (CNS)-protective activation. In contrast, Avdoshina et al. [33] observed that serum BDNF in HIV-1-positive women decreased compared to HIV-negative women, but NGF did not differ. These authors speculate that a decrease in BDNF in patients with HIV-1 could be related to increased apoptosis of T cells. They also reported that serum BDNF of HIV-positive drug users was higher than in HIV-positive non-drug users, suggesting that polydrug use could change serum BDNF levels [33]. The effect of viral infections on the levels of NFs in biological fluids is likely influenced by several factors related to the specific infectious disease and the immune system.

NGF and BDNF concentrations were not affected by the elapsed time from infection to milk collection in mothers with a confirmed COVID-19 PCR test or in mothers with viral symptoms suggestive of COVID-19. This finding could be related to variation between individual mothers for returning to the baseline counts in NFs-producing lymphocytes (or other NFs-producing cells) after COVID-19 infection. The time to return to the baseline counts in lymphocytes may depend on the severity of SARS-CoV-2 infection and factors influencing the immune system, including nutrition, health conditions, and genetic factors. Caggiula et al. [34] reported that serum BDNF increased (9-fold) from baseline to the relapse phase (~3 months after stable phase) and reduced during the post-relapse phase (~2 months from a relapse). Serum NGF was stable from baseline to relapse phage and increased (2-fold) during the post-relapse phase.

We observed that NGF and BDNF concentrations varied between individual mothers in the three groups (COVID-19 PCR, viral symptoms, and unexposed) from non-detectable to 2140 pg/mL for NGF and from non-detectable to 207 ng/mL for BDNF. Perrin et al. [35] found no detectable BDNF level in all breast milk samples from 74 healthy women in the United States (maternal age, 31–33 years; postpartum time, 7–14 months). On the other hand, Dangat et al. [36] reported that colostrum from healthy mothers (3 days of postpartum age) had 1.3-fold lower NGF and 8-fold higher BDNF than our results. Plasma NGF and BDNF levels were comparable to colostrum NGF and BDNF levels [36]. Maternal factors could influence the NF production in human milk. BDNF in human milk has also been reported to be 2-fold higher in mothers with vaginal birth than in mothers with a cesarean section. Milk BDNF was also 33-fold higher in term-delivering mothers than in preterm-delivering mothers (0.03 ng/mL) [11]. These authors reported that milk BDNF level was ~100-fold lower than the corresponding cord blood values.

In contrast to COVID-19 and unexposed groups, BDNF concentration in human milk was lower in mothers with mastitis than in mothers without mastitis. This result could be related to maternal stress, fatigue, and other neurological symptoms (anxiety and depression, and helplessness) in women experiencing mastitis [37,38]. A large cohort reported that mothers with mastitis (breast pain, milk stasis, and cracked nipples) have higher stress levels than mothers without mastitis [39]. Maternal stress can inhibit prolactin and oxytocin, resulting in a reduction in milk production [40,41]. Stress and fatigue could change the maternal immune system, making mothers more susceptible to infection and vulnerable to mastitis [42]. More studies are needed to understand the psychological impact of mastitis on human milk and infant health.

We observed that BDNF and NGF concentrations were comparable in COVID-19 PCR, viral symptoms suggestive of COVID-19, mastitis, and no mastitis groups. However, NGF was elevated compared with BDNF in the exposed COVID-19 group. Dangat et al. [36] found similar levels between BDNF and NGF in milk from mothers with preeclampsia and the healthy control group. Most studies have determined BDNF concentration in human milk or maternal serum without measuring NGF [6,7,11,43]. Interestingly, Nyárády et al. [11] reported that BDNF levels were significantly lower in preterm-delivering mothers than in term-delivering mothers. The authors [11] speculated that reduced BDNF level in preterm milk might contribute to the weaker neurodevelopment of preterm infants as BDNF is implicated in regulating neuronal growth and synaptic plasticity [3]. Serum BDNF levels were lower in preterm infants than in term infants, suggesting that BDNF serum reflects the nervous and immune systems’ maturity [7]. Whether the reduced BDNF levels in milk from mothers with mastitis influences neonatal development remains investigated.

There are few limitations in this study. First, we could not perform a survey on the pre-pandemic 2018 group to determine whether these donors had recently experienced viral symptoms in the month(s) preceding the milk collection. In both groups (COVID-19 exposed and unexposed), mothers may have been exposed to other common respiratory viruses, including influenza viruses or common human coronaviruses. However, the effect of COVID-19 infection on the secretion of NFs in human milk was evaluated in this study. A new study is needed to investigate the effect of common respiratory viruses and co-infections on human milk NFs. Secondly, mothers with viral symptoms suggestive of COVID-19 had no PCR testing. However, our recent study [44] demonstrated no significant difference in titers of IgG, secretory IgM (SIgM)/IgM, and secretory IgA (SIgA)/IgA specific to SARS-CoV-2 receptor-binding domain (RBD) between COVID-19 PCR and the viral symptoms suggestive of COVID-19 groups. The titers of RBD-specific antibodies were higher in the COVID-19 PCR group and the viral symptoms suggestive of COVID-19 than in the control group pre-pandemic [44].

## 4. Materials and Methods

### 4.1. Study Design and Participants

For the COVID-19 study, a screening survey was completed to recruit donors that had a confirmed COVID-19 PCR test. Participants were asked to report when they had a positive PCR test and their symptoms. The screening also identified donors with viral symptoms associated with COVID-19 but that did not have a PCR test. These participants also reported when they were sick and what symptoms they experienced. Milk samples collected from mothers during 2018 were used for the control pre-pandemic group. For the mastitis study, another survey was completed by donors to identify whether they had previous mastitis infections (symptoms of redness, swelling, and breast inflammation, fever, and/or intense stress and fatigue) during the last year. Donors recruited in the mastitis study were not the same donors from the COVID-19 study. The inclusion criteria were living in the USA, lactation time between 4 and 10 months, passing blood tests, and completing a health questionnaire. Written consent to use milk for research was obtained from all participants. Milk collection was approved by the institutional review board (IRB00012424) of Medolac Laboratories. Mothers who used nicotine or other narcotics were excluded from these studies.

### 4.2. Human Milk Collection

Human milk samples (150–250 mL) were collected at home with clean electric breast pumps into sterile plastic containers and stored immediately at −20 °C in deep freezers. Human milk samples were frozen and transported in insulated boxes to Medolac Laboratories, where they were kept frozen and stored at −80 °C until the ELISA measurements.

### 4.3. BDNF and NGF Concentrations

Human milk samples (2 mL) were rapidly thawed and centrifuged at 1301× *g* for 20 min at 4 °C. After removing the fat layer with cotton swabs, BDNF and NGF concentrations were determined in undiluted supernatant samples using DuoSet ELISAs (DY248 and DY256-05) and carried out as described by the manufacturer (R&D Systems Inc., Minneapolis, MN, USA). ELISAs were recorded with a microplate reader (SpectraMax iD5, Molecular Devices, Sunnyvale, CA, USA).

### 4.4. Statistical Analysis

Kruskal-Wallis test followed by Dunn’s multiple comparisons test was used to compare the measurements between COVID-19 PCR, viral symptoms, and unexposed groups using GraphPad Prism (version 9.1.0) (GraphPad Software, San Diego, CA, USA). Šídák’s multiple comparisons test was used to compare BDNF and NGF in the three groups. Linear regressions were determined between the elapsed time from infection to the milk collection in COVID-19 PCR and viral symptoms groups. Linear regressions were determined between BDNF and NGF concentrations in the three groups.

Mann–Whitney test (unpaired experimental design) was used to compare mastitis and no mastitis groups. Šídák’s multiple comparisons test was used to compare BDNF and NGF in the two groups. Linear regressions were evaluated between the elapsed time from mastitis to the milk collection in the mastitis group. Linear regressions were determined between BDNF and NGF concentrations in both groups.

The effect of maternal age and infant age on NF concentrations via linear regression were performed in COVID-19 and mastitis studies. The effect of infant gender was also determined in both investigations using Mann–Whitney tests.

The NGF averages CV across samples were 7.8 (mean) and 9.0 (median). The BDNF averages CV across samples were 5.7 (mean) and 5.5 (median). We determined the level of BDNF in half samples from mothers with COVID-19 cohort (duplicate) and half samples from unexposed mothers on one microplate, and the other haft samples from both groups were determined at the same time on another microplate. The NGF level in these samples (COVID-19 cohort) was also determined in the same experimental plan as previously described. The standard curves and blanks obtained (OD) between the two microplates of the specific measurement (BDNF or NGF) were similar. We performed the ELISAs using all samples from the mastitis cohort on two different microplates, one for NGF and another for BDNF.

No specific calculation of the sample size and power was not performed for this study. The sample size was selected based on our previous studies of sample sizes [45,46,47,48] and proved to be adequately powered based on the results. Nassar et al. 2011 [6] had a sample size of 14 infants for the breastfed group, 14 infants for the formula-fed group, and 14 infants for the mixed-fed group. These investigators found a significant difference in BDNF between the breastfed group and the formula group. Moreover, Ismail et al. 2015 [43] found significant difference in BDNF on the effect of duration of illness (≥6 months, *n* = 16 vs. ≤6 months, *n* = 14), on the effect of frequency of seizures/months (≥3 months, *n* = 13 vs. ≤3 months, *n* = 17) and the effect of disease severity (mild, *n* = 12 vs. moderate, *n* = 6 vs. severe, *n* = 12). However, a larger study is needed to confirm the effect of neurological symptoms during COVID-19 on the secretion of NFs.

## 5. Conclusions

Our study reveals that NGF concentrations in human milk were lower in mothers with a COVID-19 PCR test and mothers with viral symptoms suggesting COVID-19 than in unexposed mothers, but BDNF was comparable. The NGF and BDNF concentrations did not differ between COVID-19 PCR and viral symptoms groups. This investigation also showed that BDNF level was higher in mothers with headaches or with loss of smell/taste than in mothers without the respective symptom. BDNF concentration in human milk was lower in mothers with mastitis than in mothers without mastitis. NGF and BDNF concentrations strongly varied between individual mothers in each group. In summary, previous COVID-19 and mastitis infections changed differently the secretion of NGF and BDNF in human milk. Future studies are needed to evaluate neonatal neuronal cells’ growth when incubated with human milk samples from mothers with neurological symptoms related to viral and bacterial infections. An observational study is required to evaluate potential correlations between the human milk NF concentrations and newborns’ neurodevelopment.

## Figures and Tables

**Figure 1 ijms-22-03846-f001:**
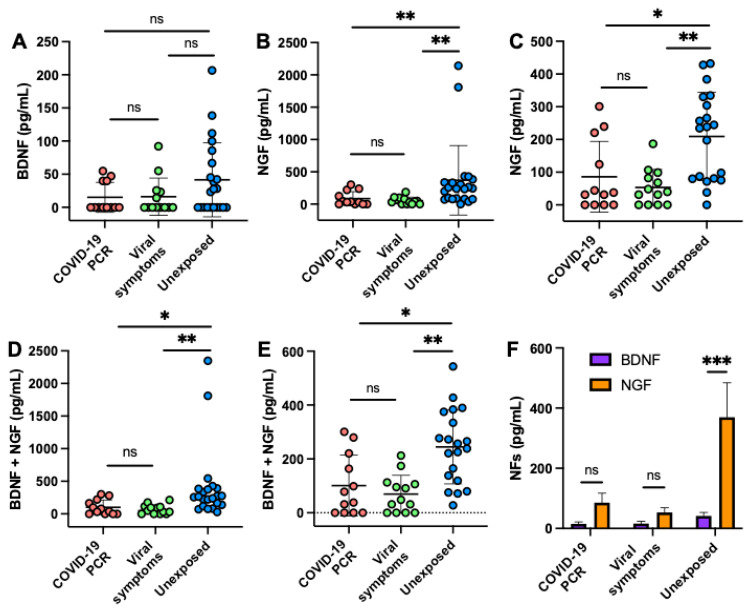
Concentration of brain-derived neurotrophic factor (BDNF) and nerve growth factor (NGF) in human milk from mothers with a confirmed COVID-19 PCR, mothers with viral symptoms suggestive of COVID-19, and unexposed mothers (control pre-pandemic 2018). Comparison of (**A**) BDNF, (**B**) NGF, (**C**) NGF (without 2 outlier cases in unexposed), (**D**) BDNF + NGF, (**E**) BDNF + NGF (without 2 outlier cases in unexposed) between COVID-19 PCR, and viral symptoms, and unexposed groups. (**F**) Comparison of NFs concentrations in human milk between BDNF and NGF in COVID-19 PCR, viral symptoms, and unexposed groups. (**A**–**E**) Values are means ± SD; (**F**) Values are means ± SEM; (**A**–**F**) *n* = 12 for mothers with a confirmed COVID-19 PCR, *n* = 13 for mothers with viral symptoms suggestive of COVID-19, and *n* = 22 for unexposed mothers. (**A**–**E**) Kruskal–Wallis test followed by Dunn’s multiple comparisons test was used to compare the three groups. (**F**) Šídák’s multiple comparisons test was used to compare BDNF and NGF in each group. Asterisk shows statistically significant differences between variables (*** *p* < 0.001; ** *p* < 0.01; * *p* < 0.05). ns, not significant.

**Figure 2 ijms-22-03846-f002:**
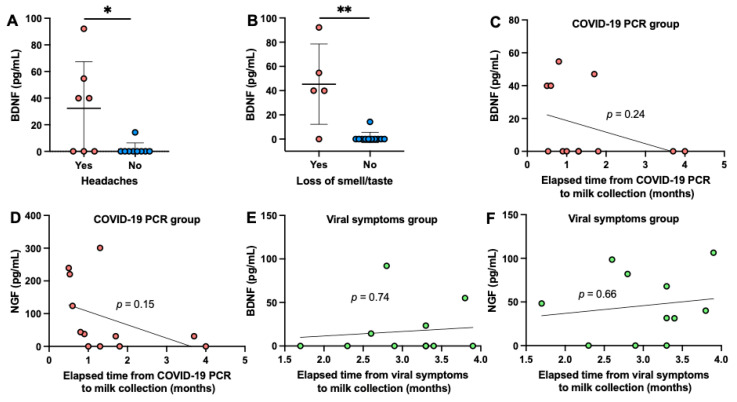
Effect of viral symptoms (likely related to the central nervous system) and elapsed time from infection to milk collection on the concentration of brain-derived neurotrophic factor (BDNF) and nerve growth factor (NGF) in human milk. (**A**) BDNF concentration in mothers with headaches (*n* = 7) and mothers without headaches (*n* = 9) during COVID-19 infection. (**B**) BDNF concentration in mothers with loss of smell/taste (*n* = 5) and mothers without loss of smell/taste (*n* = 11) during COVID-19 infection. Linear correlation between the elapsed time from COVID-19 PCR test to milk collection and (**C**) BDNF concentration or (**D**) NGF concentration in milk from mothers with a confirmed COVID-19 PCR test (*n* = 12). Linear correlation between the elapsed time from viral symptoms to milk collection and (**E**) BDNF concentration or (**F**) NGF concentration in milk from mothers with viral symptoms suggestive of COVID-19 (*n* = 13). (**A**,**B**) Mann–Whitney test was used to compare the two groups. Values are means ± SD. Asterisk shows statistically significant differences between variables (** *p* < 0.01; * *p* < 0.05).

**Figure 3 ijms-22-03846-f003:**
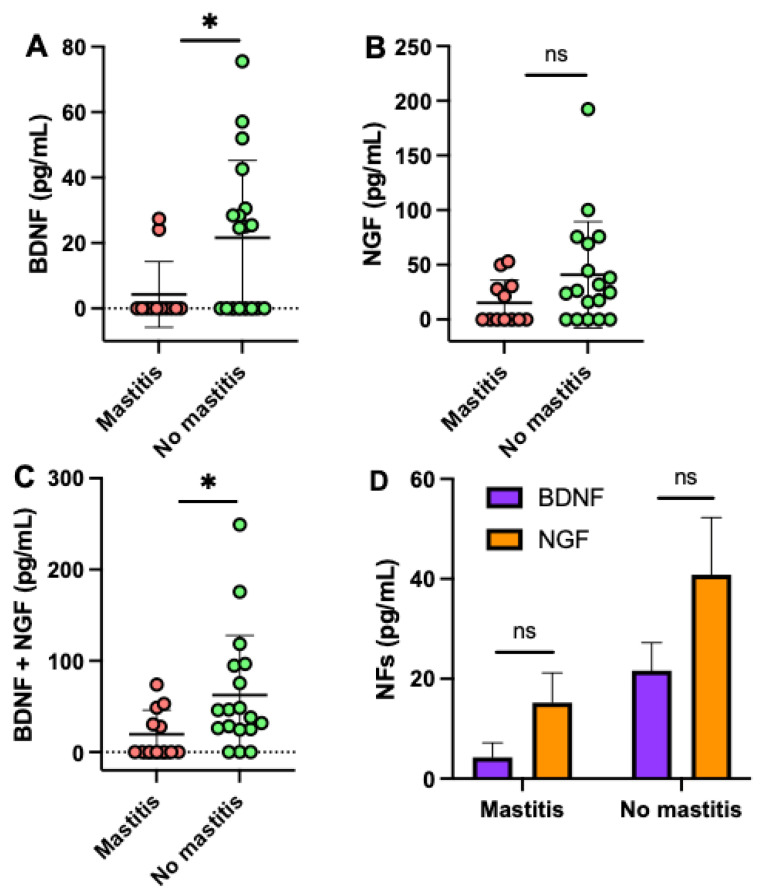
Concentration of brain-derived neurotrophic factor (BDNF) and nerve growth factor (NGF) in human milk from mothers with mastitis and mothers without mastitis. Comparison of (**A**) BDNF, (**B**) NGF, and (**C**) BDNF+NGF between mastitis (*n* = 12) and no mastitis groups (*n* = 18). (**A**–**C**) Values are means ± SD, where Mann–Whitney test was used to compare the two groups. (**D**) Values are means ± SD, where Šídák’s multiple comparisons test was used to compare BDNF and NGF in each group. Asterisk shows statistically significant differences between groups (* *p* < 0.05). ns, not significant. Women with mastitis had redness, swelling, and pain in breast, fever, and fatigue. The elapsed time from mastitis to milk collection was 1 to 4 months.

**Table 1 ijms-22-03846-t001:** Demographic description (self-reported) of mothers with confirmed COVID-19 PCR test, mothers with viral symptoms suggestive of COVID-19, and unexposed mothers (control pre-pandemic 2018).

Demographics	COVID-19 PCR (*n* = 12) ^2,3^	Viral Symptoms (*n* = 13) ^3^	Unexposed (*n* = 22)
Postpartum time, months ^1^	5 ± 2 (4−10)	6 ± 2 (4−9)	6 ± 1 (5−8)
Infant gender, *n*	6 males: 6 females	5 males: 8 females	3 males: 3 females
Maternal age, years ^1^	31 ± 4 (26−37)	32 ± 5 (23−40)	33 ± 4 (25−39)
Headaches, *n*	4	3	NA
Loss of smell and taste, *n*	4	1	NA
Fatigue, *n*	3	2	NA
Nasal congestion, *n*	0	7	NA
Date of infection	03/29/20 to 10/13/20	3/08/20 to 6/25/20	NA
Time from infection to collection, months ^1^	2 ± 1 (0.5−4)	3.0 ± 0.7 (1.7−3.9)	NA

^1^ Data are mean ± SD, min, and max; ^2^ Women were diagnosed with COVID-19 PCR test with a nasal swab (positive RNA SARS-CoV-2).; ^3^ Viral symptoms reported by COVID-19 PCR and viral symptom suggestive of COVID-19 groups (*n* = 16) were headaches, loss of taste and smell, fatigue, fever, nasal congestion, cough, severe upper respiratory infection, and body aches. Milk collection was performed after COVID-19 infection. Some women with COVID-19 PCR did not report symptoms (*n* = 6) (see Appendix A for characteristics of individual mothers). NA, not applicable.

**Table 2 ijms-22-03846-t002:** Demographic description (self-reported) of mothers with mastitis and mothers without mastitis.

Demographics	Mastitis Group (*n* = 12) ^2^	No Mastitis Group (*n* = 18)
Postpartum time, months ^1^	6 ± 2 (3−10)	6 ± 2 (3−10)
Infant gender, *n*	6 males: 6 females	10 males: 8 females
Maternal age, years ^1^	30 ± 4 (24−35)	33 ± 5 (23−40)
Stress, *n*	12	0
Fatigue, *n*	12	0
Date of mastitis(diagnosis/symptoms)	07/12/19 to 08/17/20	NA
Time from infection to collection, months ^1^	2 ± 1 (0.5−4)	NA

^1^ Data are mean ± SD, min, and max; ^2^ Women with mastitis had redness, swelling, and pain in breast (inflammation), fever, stress, and fatigue. NA, not applicable.

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
