# Peer review of "Influence of Previous COVID-19 and Mastitis Infections on the Secretion of Brain-Derived Neurotrophic Factor and Nerve Growth Factor in Human Milk"

_ijms, 2021, doi:10.3390/ijms22083846_

Round 1

Reviewer 1 Report

The authors conducted an interesting study on the influence of recent COVID19 infection, as well as mastitis, on NGF and BDNF levels of human milk. The conclusions are intriguing and can trigger future studies to specifically assess the impact of those changes in neurotrophic factors in the neuro-development of infants. 

I have a few methodological concerns, however, that I would like the authors to address:

1) How confident can one be that the unexposed 22 mothers (pre-pandemic) represent a true control group? It appears that any recent viral infection may potentially impact the secretion of NGF in human milk, therefore how could one ascertain that the 22 control mothers did not experience even minor symptoms of a viral infection (eg URI) in the month(s) preceding the milk collection?

2) From review of the graphs, it appears that the difference in NGF levels between mothers with COVID19 and control group, is largely driven by two outlier cases in the control group (Figure 1B). Do the authors believe that the statistical significance of this result would hold in a bigger study?

3) Given the known methodological problems of ELISA assays, I would like to know more details about the reproducibility of the assay they used to calculate BDNF and NGF levels. 

4) Can the authors provide more information on the calculation of the sample size and power?

Author Response

Response to Reviewer 1

Manuscript ID: ijms-1149862

Title: Influence of Previous COVID-19 Infection on the Secretion of 
Brain-derived Neurotrophic Factor and Nerve Growth Factor in Human Milk

We have made all the reviewer requested changes and hope you agree that the revised manuscript is now suitable for publication. Edited text in the revised manuscript is highlighted yellow. We have also responded to each of the reviewer comments below. We would like to take this opportunity to thank the reviewers for helping to improve our manuscript.

----

The authors conducted an interesting study on the influence of recent COVID19 infection, as well as mastitis, on NGF and BDNF levels of human milk. The conclusions are intriguing and can trigger future studies to specifically assess the impact of those changes in neurotrophic factors in the neuro-development of infants. 

I have a few methodological concerns, however, that I would like the authors to address:

How confident can one be that the unexposed 22 mothers (pre-pandemic) represent a true control group? It appears that any recent viral infection may potentially impact the secretion of NGF in human milk, therefore how could one ascertain that the 22 control mothers did not experience even minor symptoms of a viral infection (eg URI) in the month(s) preceding the milk collection?

>> Thank you for this great comment. We could not perform a survey on the pre-pandemic 2018 group to determine whether these donors had recently experienced viral symptoms in the month(s) preceding the milk collection. In both groups (COVID-19 exposed and unexposed), mothers may have been exposed to other common respiratory viruses, including influenza viruses or common human coronaviruses. However, the effect of COVID-19 infection on the secretion of NFs in human milk was evaluated in this study. A new study is needed to investigate the effect of common respiratory viruses and co-infections on human milk NFs. We added this information in the discussion (p. 8, lines 261-268).

From review of the graphs, it appears that the difference in NGF levels between mothers with COVID19 and control group, is largely driven by two outlier cases in the control group (Figure 1B). Do the authors believe that the statistical significance of this result would hold in a bigger study?

>> Thank you. We added new figures without the 2 points that were outlier cases in the control group. We found that NGF level was higher in unexposed groups than in exposed group (p < 0.001, Figure 1E). Therefore, we believe that the statistical significance of this result will hold in a larger study.

Given the known methodological problems of ELISA assays, I would like to know more details about the reproducibility of the assay they used to calculate BDNF and NGF levels. 

>> The NGF averages CV across samples were 7.8 (mean) and 9.0 (median). The BDNF averages CV across samples were 5.7 (mean) and 5.5 (median). We determined the level of BDNF in half samples from mothers with COVID-19 cohort (duplicate) and half samples from unexposed mothers on one microplate, and the other haft samples from both groups were determined at the same time on another microplate. The NGF level in these samples (COVID-19 cohort) was also determined in the same experimental plan as previously described. The standard curves and blank obtained (OD) between the two microplates of the specific measurement (BDNF or NGF) were similar. We performed the ELISAs using all samples from the mastitis cohort on two different microplates, one for NGF and another for BDNF. We added this information in the Statistical analysis (p. 9, lines 317-326).

Can the authors provide more information on the calculation of the sample size and power?

>> No specific calculation of the sample size and power was performed for this study. The sample size was selected based on previous studies of sample sizes and proved to be adequately powered based on the results. Nassar et al. 2011 had a sample size of 14 infants for breastfed group, 14 infants for formula-fed group, and 14 infants for mixed-fed group. These investigators found significant difference in BDNF between breastfed group and formula group. Moreover, Moreover, Ismail et al. 2015 found significant difference in BDNF on the effect of duration of illness (³6 months, n = 16 vs. £6 months, n = 14), on the effect of frequency of seizures/months (³3 months, n = 13 vs. £3 months, n = 17) and the effect of disease severity (mild, n = 12 vs. moderate, n = 6 vs. severe, n = 12).

-Nassar et al. 2011. Neuro-developmental outcome and brain-derived neurotrophic factor level in relation to feeding practice in early infancy. Maternal & Child Nutrition 7, 188-197. DOI: 10.1111/j.1740-8709.2010.00252.x

-Ismail et al. 2015. Brain-derived neurotrophic factor in sera of breastfed epileptic infants and in breastmilk of their mothers. Breastfeeding Medicine 10, 277-282. DOI: 10.1089/bfm.2015.0008

However, a larger study is needed to confirm the effect of neurological symptoms during COVID-19 on the secretion of NFs. We added this information in the limitation section (p. 9-10, line 326-336).

Reviewer 2 Report

Manuscript : ijms-1149862

Title:  Influence of Previous COVID-19 Infection on the Secretion of Brain-derived  Neurotrophic Factor and Nerve Growth Factor in Human Milk

Comment for the authors

 The article entitledInfluence of Previous COVID-19 Infection on the Secretion of Brain-derived  Neurotrophic Factor and Nerve Growth Factor in Human Milk”  investigated the impact  of COVID-19 infection  on BDNF and NGF concentration in human milk  samples.  The concentration of these neurotrophins  were also studied in different milk samples obtained from mothers with and without mastitis. The results showed that NGF concentration was lower in the COVID group than in controls ; BDNF was lower in mothers with mastitis than in controls.

The aim of the work is also  interesting considering  the heterogeneous data relating to the concentrations of BDNF and NFG in breast milk in the previous works. However, my major concern with this article is on  the experimental groups.

-All groups have a low number of participants .

-Mothers included in the group with "viral symptoms" do not have a clinical / instrumental diagnosis of COVID-19 infection but only symptoms reported through a questionnaire. For this reason, their data cannot be associated with those of mothers with positive COVID -19 PCR tests.

- Control groups are not adequately described. All the clinical characteristics of the participants  are missing.

  • Data concerning the study with milk from mothers with and without mastitis have no relationship with those concerning COVID-19 infection and therefore should be included in the title or be part of another work.

Author Response

Response to Reviewer 2

Manuscript ID: ijms-1149862

Title: Influence of Previous COVID-19 Infection on the Secretion of 
Brain-derived Neurotrophic Factor and Nerve Growth Factor in Human Milk

We have made all the reviewer requested changes and hope you agree that the revised manuscript is now suitable for publication. Edited text in the revised manuscript is highlighted yellow. We have also responded to each of the reviewer comments below. We would like to take this opportunity to thank the reviewers for helping to improve our manuscript.

----

The article entitled Influence of Previous COVID-19 Infection on the Secretion of Brain-derived Neurotrophic Factor and Nerve Growth Factor in Human Milk”  investigated the impact  of COVID-19 infection  on BDNF and NGF concentration in human milk  samples.  The concentration of these neurotrophins were also studied in different milk samples obtained from mothers with and without mastitis. The results showed that NGF concentration was lower in the COVID group than in controls ; BDNF was lower in mothers with mastitis than in controls.

The aim of the work is also interesting considering the heterogeneous data relating to the concentrations of BDNF and NFG in breast milk in the previous works. However, my major concern with this article is on the experimental groups.

-All groups have a low number of participants .

>> The sample size was selected based on previous studies of sample sizes and proved to be adequately powered based on the results. Nassar et al. 2011 had a sample size of 14 infants for breastfed group, 14 infants for formula-fed group, and 14 infants for mixed-fed group. These investigators found significant difference in BDNF between breastfed group and formula group. Moreover, Moreover, Ismail et al. 2015 found significant difference in BDNF on the effect of duration of illness (³6 months, n = 16 vs. £6 months, n = 14), on the effect of frequency of seizures/months (³3 months, n = 13 vs. £3 months, n = 17) and the effect of disease severity (mild, n = 12 vs. moderate, n = 6 vs. severe, n = 12).

-Nassar et al. 2011. Neuro-developmental outcome and brain-derived neurotrophic factor level in relation to feeding practice in early infancy. Maternal & Child Nutrition 7, 188-197. DOI: 10.1111/j.1740-8709.2010.00252.x

-Ismail et al. 2015. Brain-derived neurotrophic factor in sera of breastfed epileptic infants and in breastmilk of their mothers. Breastfeeding Medicine 10, 277-282. DOI: 10.1089/bfm.2015.0008

However, a larger study is needed to confirm the effect of neurological symptoms during COVID-19 on the secretion of NFs. We added this information in the limitation section (p. 9-10, line 328-336).

-Mothers included in the group with "viral symptoms" do not have a clinical / instrumental diagnosis of COVID-19 infection but only symptoms reported through a questionnaire. For this reason, their data cannot be associated with those of mothers with positive COVID -19 PCR tests.

>> Thank you for this comment. We previously determined the titers of antibodies specific to SARS-CoV-2 RBD in human milk samples between the COVID-19 PCR group and the viral symptoms suggestive of COVID-19. We found no difference in titers of IgG, IgM, and IgA between the COVID-19 PCR group and the viral symptoms. The titers of antibodies specific to SARS-CoV-2 RBD were higher in the COVID-19 PCR group and the viral symptoms than in the control group pre-pandemic. These results are presented in a manuscript recently accepted in the Journal of Pediatric Gastroenterology & Nutrition. We added this information in the discussion (p. 7, lines 268-274).

Demers-Mathieu, V.; DaPra, C.; Mathijssen, G.; Medo, E. RBD SARS-CoV-2-specific antibodies in human milk from mothers with COVID-19 PCR or with symptoms suggestive of COVID-19. J. Pediatr. Gastroenterol. Nutr. 2021 (accepted).

- Control groups are not adequately described. All the clinical characteristics of the participants are missing.

>> The unexposed control group from COVID-19 did not have viral symptoms suggestive of COVID-19 infection, date of infection and time from infection to milk collection as they are mothers from pre-pandemic COVID-19. We changed “-“ for “NA” (not applicable) in the Table 1. We also added information in the Table 2 to specific that the control group without mastitis did not have intense stress and fatigue during milk collection by adding “0” in stress and fatigue.

- Data concerning the study with milk from mothers with and without mastitis have no relationship with those concerning COVID-19 infection and therefore should be included in the title or be part of another work.

>>Thank you for this great comment. We added mastitis in the title of the manuscript.

Round 2

Reviewer 2 Report

Manuscript : ijms-1149862

Title:  Influence of Previous COVID-19  and Mastitis Infections on the Secretion of Brain-derived  Neurotrophic Factor and Nerve Growth Factor in Human Milk

Comment for the authors

Despite the authors' answers, my perplexities on the experimental groups always remain. I believe it is not correct to merge the data of the group with SARS-CoV-2 PCR test, with those “with viral symptoms”  in which the coronavirus has never been researched.

 It could be useful to add an additional group formed by participants with negative PCR tests to SARS-CoV2.

The data reported in Table 1 are not consistent with the statement "Some women with Covid-19 PCR did not report symptoms (n = 9)".

Author Response

Reviewer 2

Despite the authors' answers, my perplexities on the experimental groups always remain. I believe it is not correct to merge the data of the group with SARS-CoV-2 PCR test, with those “with viral symptoms” in which the coronavirus has never been researched.

 It could be useful to add an additional group formed by participants with negative PCR tests to SARS-CoV-2.

>>Thank you for your comment. We separated the group with symptoms from the group with SARS-CoV-2 PCR in the Figure 1-2 and made corrections across the manuscript.

The data reported in Table 1 are not consistent with the statement "Some women with Covid-19 PCR did not report symptoms (n = 9)".

>>Thank you to find this mistake. We corrected the sentence for “Some women with COVID-19 PCR did not report symptoms (n = 6). We also added the Table S1 to describe the characteristics of each individual mother in the COVID-19 PCR and viral symptoms suggestive of COVID-19 groups.